# What factors shape surgical access in West Africa? A qualitative study exploring patient and provider experiences of managing injuries in Sierra Leone

Amrit Virk ,[1] Mohamed Bella Jalloh ,[2] Songor Koedoyoma,[2] Isaac O Smalle,[3,4] William Bolton,[5] J A Scott,[5] Julia Brown,[5] David Jayne,[5] Tim Ensor ,[5] Rebecca King[5]

[1]Global Health Policy Unit, School of Social and Political Science, University of Edinburgh, Edinburgh, UK
[2]College of Medicine and Allied Health Sciences, University of Sierra Leone, Freetown, Sierra Leone
[3]Department of Surgery, College of Medicine, University of Sierra Leone, Freetown, Sierra Leone
[4]Department of Global Health, King's College, London, UK
[5]School of Medicine, University of Leeds, Leeds, UK

**Correspondence to**
Dr Amrit Virk; avirk@ed.ac.uk

## ABSTRACT

**Introduction** Surgical access is central to universalising health coverage, yet 5 billion people lack timely access to safe surgical services. Surgical need is particularly acute in post conflict settings like Sierra Leone. There is limited understanding of the barriers and opportunities at the service delivery and community levels. Focusing on fractures and wound care which constitute an enormous disease burden in Sierra Leone as a proxy for general surgical need, we examine provider and patient perceived factors impeding or facilitating surgical care in the post-Ebola context of a weakened health system.

**Methods** Across Western Area Urban (Freetown), Bo and Tonkolili districts, 60 participants were involved in 38 semistructured interviews and 22 participants in 5 focus group discussions. Respondents included surgical providers, district-level policy-makers, traditional healers and patients. Data were thematically analysed, combining deductive and inductive techniques to generate codes.

**Results** Interacting demand-side and supply-side issues affected user access to surgical services. On the demand side, high cost of care at medical facilities combined with the affordability and convenient mode of payment to the traditional health practitioners hindered access to the medical facilities. On the supply side, capacity shortages and staff motivation were challenges at facilities. Problems were compounded by patients' delaying care mainly spurred by sociocultural beliefs in traditional practice and economic factors, thereby impeding early intervention for patients with surgical need. In the absence of formal support services, the onus of first aid and frontline trauma care is borne by lay citizens.

**Conclusion** Within a resource-constrained context, supply-side strengthening need accompanying by demand-side measures involving community and traditional actors. On the supply side, non-specialists could be effectively utilised in surgical delivery. Existing human resource capacity can be enhanced through better incentives for non-physicians. Traditional provider networks can be deployed for community outreach. Developing a lay responder system for first-aid and front-

## Strengths and limitations of this study

► This study provides an in-depth understanding of the barriers and opportunities for improving surgical access in Sierra Leone as perceived by patients and providers.
► It addresses a critical knowledge gap in understanding treatment seeking and care pathways.
► It enhances knowledge of the role of traditional practitioners in surgical delivery in Sierra Leone.
► Two data collectors were practising doctors in Sierra Leone. Practising doctors might be seen as authority figures making ancillary staff and patients less likely to answer critically, given perceived power differentials in such relationships.
► This was intentionally an exploratory study, hence the transferability of the results to other contexts could be enhanced by testing patterns of surgical use and delivery with larger samples.

line support could be a useful mechanism for prompt clinical intervention.

## INTRODUCTION

Surgical access is essential for universal health coverage (UHC) which is defined as all individuals having access to good quality healthcare when needed without enduring financial hardship.[1] Yet, a staggering 5 billion people worldwide lack surgical access, especially in low-income and middle-income countries (LMICs) where 9 of 10 people lack access.[2] Overall, 11% of the global disease burden comprises conditions amenable to surgery with injuries the second leading cause of death and disability worldwide.[3] Despite this, surgical care has been a low priority for policy-makers in LMICs. Some of the least economically developed countries

in Africa are among the worst affected by the lack of surgical access. Sierra Leone, for example, has only two fully trained trauma and orthopaedic surgeons in its main tertiary hospital, despite injuries accounting for 7% of the disease burden and an estimated 325 000 deaths.[4] The need for robust surgical systems involving the provision of operative, perioperative and non-operative care, including anaesthesia in Sierra Leone and similar low-resource settings is universally accepted. There is a sound economic argument for improving access in terms of financial and productivity losses: from 2014 to 2025, over 2 million avertable fracture-associated Disability Adjusted Life Years or DALYs (number of equivalent years of life lost due to ill health, disability or untimely death) in Sierra Leone are likely. This will require an investment of US$3.4 million 'to eliminate the burden of avertable incident DALYs' through district-level hospital care.[5] Equally, death and disability from preventable conditions in LMICs reflects fundamental flaws in their health systems and poses a wider ethical challenge for equitable access.

Despite growing global recognition of the need for improving access to safe and affordable surgery, critical evidence gaps remain with limited surgical health research particularly at the health system and community levels.[6] An in-depth enquiry into surgical care seeking and delivery in Sierra Leone will provide a comprehensive account of the challenges and opportunities for strengthening surgical provision and policy design in low income countries (LICs). This will complement emerging literature on surgical need and capacity in the country.[5 6]

### Study setting
Sierra Leone is a low-income country in West Africa with a Gross Domestic Product (GDP) per capita of US$1663 in 2018 (Purchasing Power Parity constant 2017 international $).[7] With a population of 7.7 million across 16 districts, 53% of them below the national poverty line and an average life expectancy of 51 years, Sierra Leone ranks 181 of 189 countries on the UN Human Development Index.[8] The three-tiered public healthcare system includes peripheral health units, 21 district secondary hospitals, and three referral hospitals, besides 45 private clinics and 27 private hospitals, with a concentration of facilities in the capital Freetown.[9] Only a third of hospitals in Sierra Leone offer all three Bellwether procedures (caesarean section, laparotomy and open fracture management) which are indicative of a systems capability to provide essential surgical care.

Surgical need is high with recent estimates revealing that a minimum of 1·5 million individuals indicate conditions needing a surgical consultation.[10] Non-profit facilities deliver 54% of all surgeries, government providers 39.6% and private for-profit institutions 6.4%[11] Surgical care is mainly delivered by non-specialist medical officers (MO) and surgical assistant community health officers (SACHOs) and to a lesser degree by specialists, nurses and midwives.[2] While over three-quarters of the population lives within 2 hours of a surgical provider, referrals from district hospitals to specialist surgical centres in the capital Freetown could involve long distances of up to 400 km, exacerbated by poor roads and climatic conditions. Strikingly, of patients that died after accessing care at a facility, 44.8% were reportedly misdiagnosed as not needing surgery[11] indicating the low quality of available care.

### National Institute of Health Research Global Health Research Group (GHRG): Surgical Technologies
This study is a component of a large, multidisciplinary project involving social science, engineering, statistical and clinical expertise to develop low-cost surgical technology to address unmet surgical need in under-resourced settings in North East India and Sierra Leone. Embedded within the overall project, this qualitative study was designed to generate contextual information on health system capacity and local surgical need to inform the work of the other components to develop relevant clinical and engineering interventions.

### Aim
This paper aims to examine the barriers and enablers for delivering and utilising essential surgical care services in Sierra Leone, thereby generating critical evidence on health system capacity, care giving and treatment seeking practices to inform interventions for improving surgical care for populations in low-resource settings. Focusing on fractures and both trauma related and non-trauma-related wound care which constitute an enormous disease burden in Sierra Leone as a proxy for general surgical need, we examine provider and patient perceived factors that impede or facilitate surgical care in the post-Ebola context of a weakened health system. This study is the qualitative component of a larger project to develop low cost surgical technology for safely and effectively managing fractures and wounds in the country.

### METHODOLOGY
### Study design
This is an exploratory, qualitative study involving 60 respondents through 38 semistructured interviews and 5 focus group discussions (FGDs) with a total of 22 participants conducted between February and October 2019. This study was designed through a series of discussions with different disciplinary team-members to ensure relevance to the larger GHRG project goals while retaining elements to support inductive analysis. Respondents included four categories—surgical providers, district-level policy-makers, traditional healers and patients. Interviews and FGDs were triangulated with analysis of relevant national plans, academic literature and country reports to assess the existing state of surgical care in Sierra Leone. Topic guides were codeveloped for each respondent category through a series of iterations with UK and Sierra Leone based team members comprising social scientists, anthropologists, economists and surgeons. Focused on

**Table 1** Sampling frame (Semi-structured interviews (SSI))

| | | District staff | Medical providers | Patients |
|---|---|---|---|---|
| Gender | Male | 2 | 13 | 16 |
| | Female | 0 | 0 | 7 |
| Location | Bo | 1 | 5 | 6 |
| | Freetown | 0 | 6 | 14 |
| | Tonkolili | 1 | 2 | 3 |
| Occupation | Surgeon/specialist | | 3 | – |
| | Medical officer/resident | | 9 | – |
| | Medical Superintendent | | 1 | – |
| Total | | 2 | 13 | 23 |

examining health system and individual-level capacity, processes and practices, the guides were translated into Krio, back-translated for accuracy and refined after pilot testing in the field. Study methods are reported based on guidance adapted from the Consolidated Criteria for Reporting Qualitative Research (COREQ)[12] and Standards for Reporting Qualitative Research (SRQR) guidelines.[13]

### Study districts and health facilities

This study was conducted across three districts including Western Area Urban (Freetown), Bo in the Southern province and Tonkolili in the Northern Province. Forming part of Sierra Leone's total of 16 districts, Freetown and Bo are among the most populated in the country and together with Tonkolili comprise districts with high to medium surgical volumes. Three local researchers including a surgeon specialist and two junior doctors, informed site selection that was designed to reflect districts with both high and low burden of surgical activities. Across districts, a total of six surgical care facilities were purposively identified to encompass both secondary and tertiary levels of care and a mix of government-run, private and non-governmental facilities. This included three in the capital Freetown (government tertiary, non-profit tertiary and government secondary), two in Bo (government tertiary, private for-profit) and one in Tonkolili (government secondary). In addition, we interviewed two district MO in Bo and Tonkolili from the district health management teams, units of the health ministry overseeing primary care activities and implementing disease-specific vertical programmes, to get their distinct perspectives on surgical need and provisioning locally.

### Sampling

Participants were recruited through purposive and snowballing methods to include those with first-hand knowledge and experience of service provision and utilisation of surgical services (tables 1 and 2 for details). Participants were all over 18 years. A total of 5 FGDs were conducted, 2 with traditional healers (three participants per FGD) and three with ancillary staff (nurses, SACHOs, community health officers (CHOs)) in hospitals (Freetown three participants, Bo five participants, Tonkolili eight participants). We recognise that FGD participant numbers particularly for traditional healers and SACHOs were smaller than average (typically 5–8 participants). However, traditional healers are a particularly hard to reach group within this context and pragmatic choices needed to be made with their recruitment. Our decision to continue undertaking FGDs was partly due to time and human resource constraints as well as ethical approval for FGDs. Within existing time and budget limitations, FGDs were designed to represent a broad spectrum of voices and identify specific concerns of particular groups. Importantly, trained researchers enabled a suitable environment for FGDs, ensuring interaction among FGD participants was maintained rather than a conversation

**Table 2** Sampling frame (FGD)

| Gender | Location | | | Nurse Anaesthesist* | Traditional Healers† | Nurse/scrub nurse/theatre in-charge | CHO/SACHO |
|---|---|---|---|---|---|---|---|
| | Freetown | Bo | Tonkolili | | | | |
| Male | 3 | 7 | 6 | 2 | 4 | 4 | 6 |
| Female | 0 | 1 | 5 | 0 | 2 | 4 | 0 |
| Total | 3 | 8 | 11 | 2 | 6 | 8 | 6 |

*Nurse anaesthetists are advanced practice nurses administering anaesthesia for surgery or other medical procedures.
†In this context, traditional healer otherwise called folk healer is typically an unlicensed practitioner who practices the art of healing using traditional practices and herbal remedies. They may be highly trained to pursue their specialties, learning through study, observation and replication.
CHO, community health officer; FGD, focus group discussion; SACHO, surgical assistant community health officers.

between researcher and participants as is the case with interviews.

Following written informed consent from respondents, two local researchers (SK and MBJ), both medical doctors trained in qualitative techniques, independently conducted all interviews. Dyadic FGDs were conducted, led by one researcher while the other took notes. Reflective notes were shared with the core team (AV, IOS, TE and RK) and formed part of the analysis.

## Data analysis

Analysis was informed by a pragmatic approach[14] which accounted for both emergent results grounded in the data as well as their application to the study goals to generate information on health system capacity and careseeking and delivery processes.

MBJ and SK prepared English language transcripts for SSIs and FGDs, all of which were audio-recorded. Transcripts were stored and analysed using qualitative analysis software QSR NVivo V.12. Complemented by regular virtual meetings among the core team and reflective note-taking, a single researcher (AV) carried out thematic coding using a combination of deductive and inductive techniques. Given the exploratory nature of the study an inductive approach was considered suitable. Equally, time-limited funding and an established timeline for the study prompted use of deductive techniques in the initial stages of data analysis. Our deductive approach was premised on WHO's buildings blocks framework,[15] which informed initial study design and underpinned analysis of health system capacity. Accordingly, preliminary codes were developed and organised along preselected respondent categories and a priori such as causes of, resources for and management of injuries. Rereading data resulted in further expansion of codes recognising emerging patterns related to staff motivation, patient pathways and interactions between traditional and medical systems. Using the three-delays framework,[16] we explored obstacles to timely and safe surgical care at the community and health facility levels. Through a series of iterations, generated codes were refined and categorised into higher order themes that are reflected in the discussion section.

## Patient and public involvement statement

Patients and/or the public were not involved in the design, or conduct, or reporting of this research due to unintended delays and limited time for the study completion. We would consider greater public involvement in the dissemination plans from this research.

## RESULTS
## Respondents' profile

Respondents included 46 men and 14 women, 24 located in Freetown, 20 in Bo and 18 in Tonkolili. Of 23 patients, six involved wounds/ulcers, eight lower limb fractures, one ORIF fracture, four below-knee amputations, one burn injury, one both a wound and fracture and one

unknown. In keeping with known trends for those trauma patients,[3] most patients (14) in our sample were younger (between 30 and 45 years) with more men (16) than women (7). Most medical providers we approached were employed at government hospitals, with two practising at an non governmental organisation (NGO)-run hospital in Freetown and two in a private district hospital. They were drawn from across the care spectrum to include 5 surgeons, 11 MO/Residents, 3 anaesthesists, 8 nurse/theatre in-charge and 8 CHO/SACHOs (table 2). Traditional healers were initially approached through the Sierra Leone Indigenous Traditional Healers Union and included both registered and non-registered healers, a total of four men and two women. Healers were locally trained typically through an apprenticeship model with individuals principally using plant-based medicines and specialising in managing fractures, wounds and conditions such as epilepsy.

## Demand-side challenges
### Deferral and delays in initial decision to seek treatment

As compared with patients with trauma, providers reported long delays and a more circuitous route to hospitals among patients needing non-trauma wound care who often initially self-treated or consulted local healers. Besides perceived low severity of their condition, social networks influenced such patients' care seeking pathways notably advice on visiting traditional and local practitioners. A patient in Freetown illustrated some of their reasons for deferring treatment,

> My foot started swelling... I thought it was nothing so I stayed home until two weeks without going to the hospital... a nurse...said I should go to her for injection... I saw no improvement... I even offered two goat sacrifices...but it still didn't improve. Then a man...said go to the doctor's hospital. (PT1, Female, 40–45 years, Freetown)

### Deterrents to seeking medical care

While some patients mentioned travel costs as a deterrent to timely treatment seeking, a more prominent factor was the widespread belief within communities in non-Western/medical causes of injuries, which were instead attributed to 'black magic or spiritual cause' (PT005). Accordingly, patients were impelled to consult traditional healers for considerable periods ranging from a few weeks to several years before presenting with worsened symptoms.

> It's been four years since the incident... whenever I heat the wound, it will produce some small white worms ...I have never gone to the hospital (until now). I was told (it) was a result of a fetish attack set by my enemies...I do not have money. Transportation from where I stay to the hospital I cannot afford (PT22, 40–45 years, Male, Bo)

Few patients reported distance and transport costs as a challenge or deterrent to care seeking, although treatment costs clearly factored into where they accessed care from. Patients were willing to travel longer distances to obtain treatment at facilities which were seemingly cheaper; a point illustrated by the following health worker at a large referral facility in Freetown who had earlier worked in an NGO run provincial hospital,

> … what (patients) tell me is that it is costly here. When we were in (the NGO run) hospital, the patients would tell you that it is costly…at (government hospital) and that is why some of them would travel all the way… (RB, FGD1, Govt tertiary hospital, Freetown)

Just as hospital fees deterred some patients from accessing treatment, costs of care at public hospitals and the financial strain of needing to pay out of pocket also interrupted follow-up and treatment for some.

> …theatre costs, medications we have to pay for everything…my medications…to buy, my family had to borrow money because I've missed some injections… it's not affordable; the cost is too expensive for us. (PT004, 18–25 year male, Freetown)

### Pull factors for use of traditional healers

Traditional healers indicated flexibility concerning how they were paid. While accepting payments in-kind such as livestock or household items, others moulded treatment to fit in with patient budgets as the following healer reported when asked about the affordability of treatment.

> All depends on the type of case…Some would pay higher; others would pay less… sometimes we ask the patient what they can afford and…we make a decision to treat them or not. We negotiate to a common ground. (FGD3, Healer 2, Bo)

Besides personal faith and affordability, responses highlighted patients' fears related to possible amputations in hospitals driving them towards non-medical native healers. Yet, some formal providers contended that traditional practitioners capitalised on patients' vulnerability and persuaded them to bypass hospitals, often making amputations inevitable.

> … lot of people are poor …they try the herbal medications…they come in a situation where the wounds are very bad that some of them we cannot save the limbs without amputation…most of them come very late. (MP5, Surgeon specialist, Bo)

Figure 1 illustrates key and perceived barriers to timely treatment seeking by patients.

### Front-line care: role of social networks

For gravely injured victims of road traffic and workplace accidents the first point of call was typically a hospital, with bystanders or colleagues often dictating the choice of facility. Patients with trauma were generally walk-in patients reporting self-referral to tertiary level facilities which signals limited mechanisms for gate-keeping via lower level facilities. In contrast, patients with wounds presenting at hospitals were reportedly from provinces with poor first level treatment facilities, often having delayed treatment.

> Although prompt transfer to hospitals following accidents meant fewer delays reaching medical facilities, most cases involved privately hired vehicles not equipped for first-aid or communication with hospitals. On-site first responder care was self-administered or dispensed by onlookers involving rudimentary methods using 'cloth… wooden frames' for limb stabilisation (PT9), ice-packs on fractures (PT5), "raw eggs and dust" on burns (PT16) and fuel to contain bleeding (PT6, 30–35 year Male, Bo).

| Affordability & convenience | • High out of pocket costs at government facilities<br>• Flexible payment modes of traditional healers |
|---|---|
| Personal beliefs | • Community faith in traditional methods<br>• Patients' perceived low severity of condition<br>• Patients' fear of amputations in hospitals |
| Organisational | • Limited surgical capacity at peripheral health facilities |

**Figure 1** Key barriers to timely care seeking.

Trauma site→first aid by lay responders→secondary or tertiary hospital→treatment

↓

Referral or refusal→
(*treatment cost or fixed patient criteria*)

Early signs of wounds→ peripheral heath facilities→ secondary & tertiary hospitals

Early signs of wounds→ self-treatment→ local healers→ tertiary facility

**Figure 2** Complex patient pathways to care.

Patients overwhelmingly mentioned the role of family and social networks in facilitating treatment access by arranging transport to facilities, managing admission processes and consenting to treatment. Highlighting the role of 'good Samaritan' onlookers in managing their access to treatment, a trauma victim recounted their experience shortly after they were in a road traffic accident,

> …one young boy, a bike rider… insisted that I should be brought to hospital… he went back and informed my relatives… he came with them back to hospital. I was attended to that same day—they gave me a bed and I slept. (PT002, Female, 30–35 years, Freetown)

Figure 2 illustrates some of the patient pathways to care commonly reported by our sample. This is designed to demonstrate the complex routes by which patients reach medical facilities in Sierra Leone, rather than a comprehensive account of these pathways.

### 'Carpenters without tools': Supply side factors impeding surgical treatment

Inadequate supplies and stock outs were common within all government facilities resulting in patients being asked to purchase medicines and materials from outside. Reflecting on the unintended effects of this, a surgeon in a provincial government hospital explained how having patients buy drugs made it difficult to evaluate their 'efficiency and potency' (MP13, male surgeon, Bo). Another government doctor similarly reflected on the public system's lack of essential supplies and the unintended effects of diminished patient trust in inducing them into using traditional healers.

> Sometimes we have to ask the patients to get these materials from outside including antibiotics … the patients would lose trust in our system…seeking alternative care from traditional healers which may end up killing them. (MP9, MO, Government facility Bo)

Medical providers also affirmed the well-recognised dearth of surgical specialists within public facilities and noted the role of general surgeons and junior doctors in treating most surgical cases. Shortfalls in trained staff were generally noted although non-physicians, notably nurses, SACHOs and volunteers, played a critical role as first responders and for follow-up care. This included managing conservative injuries including wound cleaning, dressing and plastering. Particularly within provincial settings, nurses and CHOs signalled their dissatisfaction with the lack of opportunities to use their knowledge including for continuous skill development. Strikingly, for ancillary staff, a lack of investment in skill development was a strong demotivating factor, as a government nurse anaesthetist in Tonkolili summed up:

> …even the instruments for practice we do not have… we are losing our skills. We want to be called on regular basis for trainings… also to be upgrading. This man has been working for the past 15 years and has not been paid salary…They are discouraging people from practicing. (No.5, FGD 4, Tonkolili)

Providers also reported limited diagnostic capability at government hospitals both in terms of the tests available and their quality. X-ray/radiology was generally available but subject to recurrent power outages. More advanced microbiology testing was outsourced to private laboratories as some district-level providers acknowledged the unreliability of in-house test results. Relatedly, some patients reported being admitted into state-run peripheral health units for up to a fortnight before being referred because of the facility's lack of expertise: 'they said they couldn't handle my problem; it had gotten to a stage beyond them….' (PT3, Female, 50–55 years, Freetown). On their part, providers also highlighted the low financial status of most of their patients who could not afford private testing further limiting providers' capacity for accurate diagnosis, increasing the risk of poor health outcomes.

> … we may want …swabs and send them for histology but the financial status for some of our patients are weak so we can't…know what the main cause is… we put them on dressing…Some can survive… and some will not. (MP15, MO, Tonkolili)

Besides specialist services like physiotherapy and prosthetics for which referrals were common, district-level providers at secondary and tertiary facilities also reported referrals to NGO-run or government specialist centres

for procedures for which they were ill equipped. These providers contended that their inability to provide certain procedures stemmed not from of lack of skill but because of deficient supplies that had led to the discontinuation of some services.

> In the past we used to do it but we have lost the equipments. We had the machine and the artificial scraper…the electrical equipment…but we have lost it now. At this time we don't do skin grafts. We would treat the wound and when it's clean we can refer them to other facility (R2, FGD2, Bo)

Providers at private facilities with access to supplies reported fewer challenges in dispensing treatment, contrasting this with lower quality of service provisioning at less equipped facilities.

> We are provided with the medications that we need and the staff are all trained and…know what to do. At (the government facility) however, you…write (prescribe) medications …realise that the patient missed a day of antibiotics which you will have to compensate for because you might start to see deteriorating condition in patient…So yes there is a lack of attention when it comes to wound care in other places. (MP3, MO, NGO tertiary hospital, Freetown)

### Delayed care following contact with medical facilities

Care pathways were not straightforward and for some involved consultations with traditional healers following visits to government hospitals, motivated by myriad factors including the burden of follow-up treatment costs, dissatisfaction with diagnosis and low levels of awareness of pre-existing conditions like diabetes in slowing recovery rates. Patients also accessed peripheral heath facilities, the first point of care for surgical patients, but both providers and patients indicated limited surgical capacity delayed treatment and presenting with worsened conditions.

> …some people…in the community…health units… not so au fait with the wound and how to take care … the health unit workers will only clean the top and not deep into the wound to remove all the dead tissues. So… we would have to clean it all over again (I3, FGD4, Tonkolili)

Patients experiencing delays reflected on the non-emergency cases, because of resource shortages within public facilities.

> … many of us are waiting to be taken to theatre but they said they ran out of stock for some drugs in theatre, so they can only take emergency cases for operation. (PT4, Male, 18–25 years, Freetown)

For others, despite reaching secondary and tertiary facilities, some patients reported delayed care and being turned away because of facilities' inability to treat certain cases due to space shortages or facility instituted admission criteria to control patient numbers. For example, a patient who sustained injuries at a construction site was diverted from an NGO-run tertiary facility in Freetown which only accepted patients with trauma 14 years and below (PT4, 18–25 years male). Reported delays also occurred because of patients' limited financial means which motivated some providers to refer them to NGO-run facilities that offered cheaper or free-of-cost services. Yet, a doctor at one such NGO-run tertiary centre acknowledged their inability to service most patients, inadvertently compelling users to seek more affordable treatment options locally.

> … we refer because of bed capacity… where (patients) have to pay for their procedures…they decide to go the traditional way… they have no choice (be)cause they cannot afford it…we actually meet the needs of 10%, the others they cannot afford what is needed so they tend to go find other alternatives. (MP3, MO, NGO tertiary hospital, Freetown)

### Strategies for improving surgical delivery

District officials and providers agreed on the need for greater government commitment towards surgically treatable ailments with clear ideas for strengthening surgical provisioning. Some favoured a role for lower-tier facilities as well as using junior staff as a means to expand surgical services. Referring to national level efforts to create a cadre of community-level surgical assistants (SACHOs), district planners felt it was important to position them within lesser equipped primary and community health centres rather than hospitals. Medical providers also mentioned more orthopaedic and wound management training for nurses alongside training triage nurses to 'identify the red signs in acutely injured patients…to identify emergencies' and to minimise seeing patients 'on the basis of when they come and not based on the severity' (MP16, MO, Freetown). Traditional healers also recognised their limitations in treating patients with severely damaged legs and reported referring such patients to hospitals. Reiterating the deep entrenchment of traditional healers locally, a district administrator proposed that they be better integrated into the formal healthcare system.

> … in Koinadugu when I was working… We built place for (traditional healers) within the hospital… when they approach things that they do not understand they will just refer directly to us… need to… train them to be referring these cases. (PL4, District MO, Tonkolili)

Taking a broader public health and development view of surgical need, a district administrator stressed the importance of multisectoral action and better enforcement of road safety rules to minimise the risk of injuries.

> Let me start with the aspect of infection which is under SLRA (Sierra Leone Road Authority) the road safety. I know they have their rules and regulations, however, enforcement of those rules and regulations

is a big issue…That is where we must start (PL3, District MO, Bo)

## DISCUSSION
Despite growing awareness of surgical access as a critical and cost-effective component of UHC, surgical utilisation in low-resource settings is inhibited by numerous factors, with limited understanding of these barriers, their complex interplay and impact on patient health. We explored experiences of treatment seeking, receiving and caregiving to understand from users and providers' perspectives the factors that inhibit and assist access to safe and timely surgical care in Sierra Leone. We consider the main lessons from this study by reviewing the supply-side and demand-side factors to facilitate policy action.

### Supply-side challenges
#### Capacity constraints
Results reveal barriers to surgical access at the system and individual levels. At the health system level, capacity shortages within facilities negatively affected when patients obtained trauma care. Unsurprisingly for a low-resource setting and as noted across sub-Saharan Africa (SSA),[2] government providers reported massive shortages of equipment, trained personnel and diagnostic capabilities. Private nonprofit facilities deliver most surgeries (54%) and perform 96% of all operative fracture treatment and 86% of all orthopaedic operations in Sierra Leone.[6] Capacity constraints in terms of equipment were particularly acute for provincial facilities that despite the necessary staff skills were sometimes ill equipped to handle certain procedures leading to avoidable referrals. Health ministry documents also affirm vast urban–rural inequities with less than a quarter of surgical providers and only 9% of specialists working in rural Sierra Leone.[9] While the extent of avoidable referrals was unknown, the results imply delayed care for some patients as well as further strains on already overburdened specialist, tertiary facilities. Increasing the staffing and equipment capacity of district facilities in SSA is proposed as a key strategy for improving front-line care especially in rural areas.[17 18] However, within a general context of few resources, some studies note the increasing relevance of low-cost surgical innovations in facilitating surgical delivery in LMICs, such as mosquito nets for hernia repair.[3 19]

#### Staff motivation and incentives
Relatedly, outside the capital Freetown, much like South and East Africa,[20] non-physicians and particularly non-specialists are critical providers of surgical care in Sierra Leone.[21] Yet as recent studies[22] highlight, low staff motivation and further training are important concerns for district-level nurse volunteers and CHOs. Universally, career development opportunities offer a powerful non-financial incentive for improved health worker motivation.[23] Knowing how curbed prospects have contributed to high losses of CHOs to private facilities in Sierra Leone,

there is a strong case for policymakers to institute accreditation and governance structures for continued staff development and retention. Equally, systems for regular and objective monitoring and supervision of SACHOs' clinical performance will need to be instituted to ensure safe and effective surgical care. Ongoing professional development and training for non-physician cadres offers a promising strategy within a wider country context of limited funds and competing health priorities.

### Limited front-line support
At the individual patient level, trauma and front-line support was minimal, with indiscriminate first aid practices prevalent. Moreover, there was an overwhelming reliance on privately hired vehicles such as taxis and motorbikes for transporting patients to hospitals. While there are safety concerns associated with inadequate first aid for victims through ill-equipped commercial vehicles and lay responders, the feasibility and costs of scaled up state-run or ambulance emergency services is considered unlikely in low-resource settings.[24 25] Yet, the evident onus of prehospital care borne by citizens offers an opportunity for developing a lay responder system as recommended by WHO and tested in LMICs including Ghana, Madagascar, Iraq and Uganda.[23] This would involve training lay responders like commercial drivers, police and community leaders to deliver basic first aid and prompt transfer to hospitals.

### Demand-side challenges
#### Patients' health beliefs
Simultaneously, cultural and structural impediments at patient level need to be addressed. Poor understanding of when to seek medical help and the overwhelming influence of cultural beliefs about the causes of ill health impacted patients' care pathways delaying early medical intervention. One aspect of the policy response will clearly involve greater community-level outreach to restore public trust, deeply eroded post-Ebola.[26] However, interventions and policies will need to be cognizant of deep-rooted faith in traditional providers and mechanisms to use and engage their networks for awareness raising within communities and to incentivise earlier treatment seeking and patient adherence may be explored. Despite well-known challenges in integrating traditional and medical practices as noted in other studies in Africa,[27 28] initial steps could involve working with existing bodies such as the Sierra Leone Traditional Healers Union to identify stakeholders and roles.

#### Affordability of care
Like elsewhere in SSA, out-of-pocket costs emerged as a prominent barrier against timely treatment and adherence (figure 3 for demand–supply–payer interactions). Major surgery in Sierra Leone involves an average out-of-pocket cost of US$117, prohibitive when considered against a per capita annual income of US$490. With around 60% of the population below the national poverty

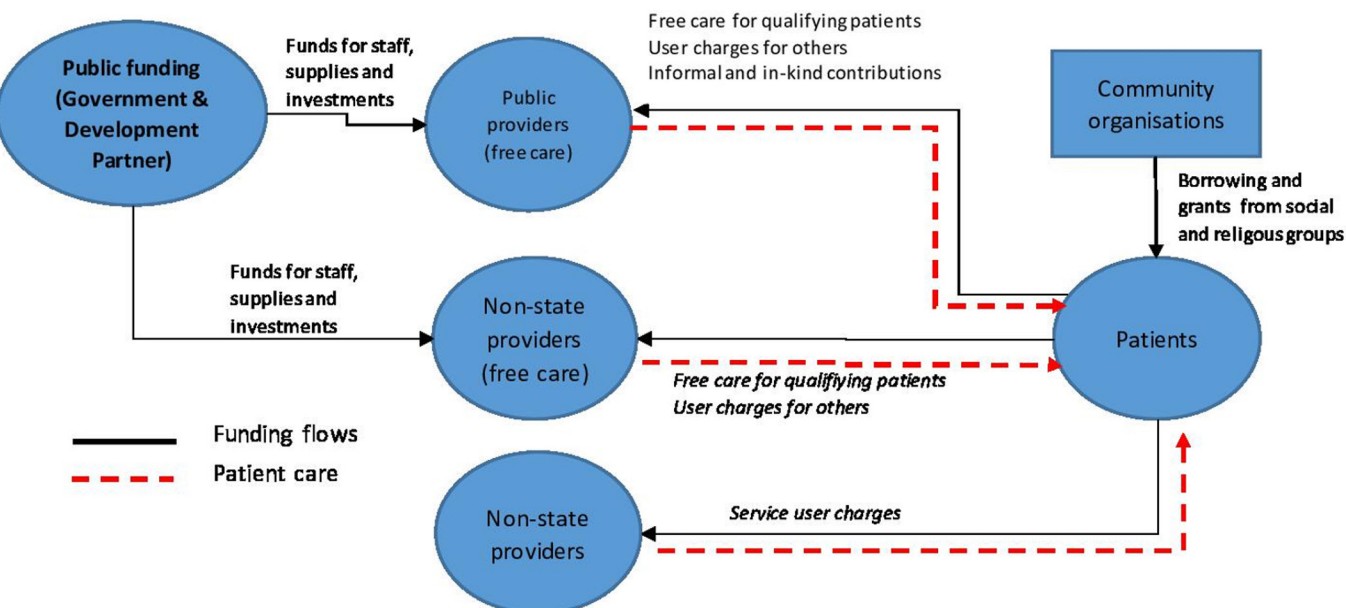

**Figure 3** Demand–supply–payer interaction for surgical services.

line, an estimated 73% of Sierra Leoneans are at risk of further impoverishment if undergoing surgery.[29]

Mechanisms for increasing the currently low levels of 2% of the population covered by health insurance urgently needs to be a policy priority. Complementary measures for partial financial support like vouchers, subcontracts with transport bodies, and community loans that have been tried for other health problems elsewhere[30] may also be considered in Sierra Leone. Based on research across Guinea, Madagascar and the Republic of Congo, Shrime *et al*[30] demonstrated improvement in surgical uptake by patients when transportation costs were paid for. They argue that transport costs are often overlooked as non-medical costs but could constitute up to 30% of patient expenditure. Considering the reported lack of surgical capacity at district level, this may be particularly important for rural patients needing essential surgery.

Strikingly, as some district officials noted and a recent high-level report[9] to the health ministry emphasises, Sierra Leone's low level of surgical provision is addressable. Strengthening health systems and surgical capacity could address critical conditions like appendicitis, obstructed labour, hernia, congenital anomalies and breast and cervical cancer. Additionally, a window of opportunity exists for legislative, regulatory and community action to prevent and control the contextual causes of surgical problems. Instituting or enforcing road safety rules is a promising strategy for minimising the risk of road traffic accidents and needs to be integrated into national public health planning. Complemented by facility-level institution of clear protocols and strengthening, this could prevent treatment delays and improve surgery-related morbidity and mortality in the country.

These findings resonate with the challenges to surgical care identified in other studies.[3 30] This includes individual level barriers at the patient and provider levels as well as institutional and systemic challenges. Key lessons from this study will be pertinent to other low-resource settings. This includes, on the demand side, financial mechanisms for ensuring more affordable care for patients as well as systematic community outreach to improve awareness about early care seeking. On the service delivery side, our results highlight the importance of continuing professional development for incentivising surgical providers and strengthening communication systems between facilities to reduce treatment delays.

## Limitations

The preceding analysis must be seen in light of the limitations of this study. The nature of the research design means that correlations between factors and differences in responses based on demographic characteristics could not be established without accompanying quantitative analysis. The size of the focus groups with traditional healers and surgical assistants was smaller than the average number of 6–8 participants in FGDs. Some guidance on FGD composition indicates that taking the focus group as the unit of analysis, the total number of groups can be increased to make up for smaller group sizes.[31] Bearing in mind mixed views on the appropriate size for FGDs,[32 33] we tried to ensure that focus groups were broadly representative of a range of perspectives on surgical delivery including different levels of seniority, roles (traditional, medical) and occupational categories.

## CONCLUSION

Injuries are already the second biggest cause of disease worldwide and with increasing urbanisation, traffic-related and industrial accidents the need for trauma-related surgery will continue to be important. Assessing the challenges and barriers to the utilisation and delivery

of emergency and essential surgical care generates insights into the levers and opportunities for improving the access, affordability and availability of surgical care in Sierra Leone. Within a context of constrained resources, lower tier staff including nurses and non-specialists could be effectively used for surgical diagnosis and delivery. With suitable in-house training and supportive supervision, a cadre of surgical assistants deployed at primary and community-level facilities would aid the expansion of surgical provisioning in Sierra Leone. Together with strengthening provincial health facilities, better integrated referral systems using community and traditional networks could be explored. Given Sierra Leone's ongoing cost recovery scheme (2006) that introduced user fees to generate revenues for public health facilities, developing appropriate financing mechanisms to curb high out-of-pocket expenses is likely to prove more challenging. Strong policy support demonstrated through committed funding and resources will need to be backed up with supportive legislation which integrates surgical care into national planning exercises.

**Contributors** All authors, AV, MBJ, SK, IOS, WB, JAS, JB, DJ, TE and RK, were involved in the conceptualisation and planning of the study. AV, IOS, MBJ and SK were involved with conducting the study, with data collection. AV, MBJ, SK, IOS, WB, JAS, JB, DJ, TE and RK were involved with the analysis and interpretation of data. AV prepared the first draft of the manuscript. All authors, AV, MBJ, SK, IOS, WB, JAS, JB, DJ, TE and RK, contributed to subsequent revisions to the manuscript. All authors, AV, MBJ, SK, IOS, WB, JAS, JB, DJ, TE and RK, read and approved the final manuscript.

**Funding** This work is supported by the National Institute of Health Research (NIHR) Grant number 16/137/44.

**Competing interests** None declared.

**Patient consent for publication** Not required.

**Ethics approval** Ethical approval was obtained from the Sierra Leone Ethics and Scientific Review Committee and the University of Leeds School of Medicine Research Ethics Committee.

**Provenance and peer review** Not commissioned; externally peer reviewed.

**Data availability statement** No data are available. As per the ethical conditions for this study, participants of this study did not agree for their data to be shared publicly, so supporting data are not available.

**ORCID iDs**
Amrit Virk http://orcid.org/0000-0001-8686-2776
Mohamed Bella Jalloh http://orcid.org/0000-0002-3172-2749
Tim Ensor http://orcid.org/0000-0003-0279-9576

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
