## [Reviewer comments · BMJ Open]

ARTICLE DETAILS

TITLE (PROVISIONAL)	What factors shape surgical access in West Africa? A qualitative study exploring patient and provider experiences of managing injuries in Sierra Leone
AUTHORS	Virk, Amrit; Bella Jalloh, Mohd; Koedoyoma, Songor; Smalle, Isaac; Bolton, William; Scott, DJA; Brown, Julia; Jayne, David; Ensor, Tim; King, Rebecca

VERSION 1 – REVIEW

REVIEWER	A.D Mwaka College of Health Sciences, Makerere University, Uganda
REVIEW RETURNED	04-Aug-2020

GENERAL COMMENTS	Dear Authors, This is a very interesting manuscript that aims to describe the issues related to access to quality surgical care in a post conflict and post Ebola epidemic country in sub Saharan Africa where biomedical care is faced with myriads of challenges. Results from this study can inform the planning and provisions of surgical services in the country and other countries with similar socio-economic background. I strongly feel that the following comments could help you improve the quality of the manuscript and make it more accessible to a larger readership. Abstract: Methods; Were the FGDs formed by age and sex for example. How many participants were there in each FGD group? Perhaps there were 5 (22.5) participants per group. These are about few compared to recommended number per group. This needs to come out as limitation; clearly stating how it is a limitation. Results; could the themes and subthemes be clearly outlined with examples of issues under each theme? This would certainly improve comprehension. Conclusion: should be seen to be arising from the presented results so that even if someone reads only the abstract, they can see clearly how the conclusion connects with the results. This does not seem to be the case in the current version of the manuscript. Methodology Page 8; line 10: why use both deductive and inductive approaches? What are the advantages of using both (please state the additional value gained by using both approaches)? On what was the deductive approach premised? Page 8; line 20 and 24: please provide citations for the WHO building blocks framework and three-delay framework. Patient and public involvement
---

	It may be necessary to present an argument why it was not necessary to involve these key stakeholders in the current study. Table 1 (a) could be made more informative by rotating it by 90 degrees to make the rows columns and the variables forming the columns make rows. Similarly, table 1 (b) could be improved; for example gender could form the columns and all else appear on the rows. Results  1. The enablers do not feature prominently in the results. 2. Page 8: experiences of patients' initial decision to seek treatment. Can the subthemes be presented in such a manner that they pick out and clearly show exactly what the experiences being referred to, and what about the experiences that the reader need to grasp? Everything could be experiences, but what about them that are of importance in a given context? Discussion The themes could appear at the opening of each paragraph, pointing out what is being talked about, relating them to the results of this study and to findings from other studies. This does not feature out prominently in the current version of the manuscript. Linking the discussions to the results is rather difficult. Conclusion Lines 31 to 38 seem material for the background/introduction. The concept of task-sharing seems to be appearing for the first time in line 40. It is important to ensure that enough materials exist in the result and discussions to support its appearance in the conclusion. It is however difficult to see these evidence in the current version of the manuscript. Good luck Amos
--	---

REVIEWER	James X. Zhang The University of Chicago USA
REVIEW RETURNED	10-Aug-2020

GENERAL COMMENTS	This is an exploratory, qualitative study investigating the determinants of surgical access in a low-income country in West Africa. A better understanding of provider- and patient-perceived factors that impede or facilitate surgical care can lead to the identification of the barriers to an equitable health care system.  1. Conceptualization. While the authors provided a detailed account of experiences of patients' initial decision to seek treatment (Page 8-10), a figure illustrating the key barriers would help conceptualize and streamline the perceived barriers, for example, monetary, belief, organizational, etc, with more detailed breakdown by each category. 2. Treatment pathways. A figure depicting/summarizing the treatment pathways (Page 10-13) can help readers to grasp the challenges in surgical care. 3. Strategies. It is unclear what the main lesson is from this study. Supply-side and demand-side factors should be discussed in a way to facilitate policy actions.
---

VERSION 1 – AUTHOR RESPONSE

Reviewer: 1

Reviewer Name: A.D Mwaka

Institution and Country: College of Health Sciences, Makerere University, Uganda Please state any competing interests or state 'None declared': None Declared

Abstract:

4. Methods; Were the FGDs formed by age and sex for example. How many participants were there in each FGD group? Perhaps there were 5 (22.5) participants per group. These are about few compared to recommended number per group. This needs to come out as limitation; clearly stating how it is a limitation.

Response: On page 6, we have now included information on the number of participants per FGD. We also identify this as a limitation and further clarify that "While participant numbers per group were small, FGD groups were mixed for age and gender, to ensure a broad spectrum of voices and specific concerns were represented."

5. Results; could the themes and subthemes be clearly outlined with examples of issues under each theme? This would certainly improve comprehension.

Response: We have reorganised the results in the abstract along the themes of demand-side and supply-side factors.

6. Conclusion: should be seen to be arising from the presented results so that even if someone reads only the abstract, they can see clearly how the conclusion connects with the results. This does not seem to be the case in the current version of the manuscript.

Response: We have reorganised the conclusion in the abstract to mirror the structure of the results section and reflect the demand and supply side thematic focus.

7. Methodology, Page 8; line 10: why use both deductive and inductive approaches? What are the advantages of using both (please state the additional value gained by using both approaches)?

Response: We have further explained the use of both deductive and inductive approaches (pages 6-7) under "Data analysis". We begin by explaining use of a pragmatic approach that "accounted for both emergent results grounded in the data as well as their application to the study goals". We also clarify the reasons for using both approaches as we go on to explain the use of an inductive approach given the exploratory nature of the study and the time-limited funding and timeline for the study directing our use of deductive techniques in the initial stages of data analysis.

8. On what was the deductive approach premised?

Response: We have now explained this on page 7 under "Data analysis" to clarify that "The deductive approach was premised on initial codes being developed and organised along pre-selected respondent categories and interview guide topics such as causes of, resources for, and management of injuries and further developed through a series of iterations to consider staff motivation, patient adherence, satisfaction, and impact."

9. Page 8; line 20 and 24: please provide citations for the WHO building blocks framework and three-delay framework.

Response We have provided the citations for the WHO building blocks framework and three-delay framework.

10. Patient and public involvement: It may be necessary to present an argument why it was not

necessary to involve these key stakeholders in the current study.

Response: We have now modified the patient and public involvement statement to explain why patients and/or the public were not involved due to unintended delays and limited time for the study completion. We further clarify that we would consider greater public involvement in the dissemination plans from this research.

11. Table 1 (a) could be made more informative by rotating it by 90 degrees to make the rows columns and the variables forming the columns make rows.

Response: We have revised table 1(a) by rotating it by 90 degrees to make the rows into columns and the variables forming the columns made into rows.

12. Similarly, table 1 (b) could be improved; for example gender could form the columns and all else appear on the rows.

Response: We have revised table 1(b) to have disaggregate data by gender, as advised.

Results

13. The enablers do not feature prominently in the results.

Response: On pg 15, (Section titled "Carpenters without tools": Surgical workforce, supplies and capacity) we have inserted an additional quote to contrast the nature of service provisioning in less and better equipped facilities to underscore the importance of provider capacity in enabling better care. This provides a link and lead into the following section (pg. 15) where we explicitly discuss strategies for improving surgical delivery.

14. Page 8: experiences of patients' initial decision to seek treatment.

Can the subthemes be presented in such a manner that they pick out and clearly show exactly what the experiences being referred to, and what about the experiences that the reader need to grasp? Everything could be experiences, but what about them that are of importance in a given context?

Response: As advised, we have reordered the text (pages 9-11) and inserted sub-headings to present additional subthemes related to the patients' experiences and initial decision to seek treatment.

15. Discussion: The themes could appear at the opening of each paragraph, pointing out what is being talked about, relating them to the results of this study and to findings from other studies. This does not feature out prominently in the current version of the manuscript. Linking the discussions to the results is rather difficult.

Response: As suggested by both reviewers (no 19 below), we have re-ordered the discussion section to have themes appear at the opening of each paragraph, pointing out what is being talked about, relating them to the results of this study, and to findings from other studies. We have organised the main lessons from this study by reviewing the supply-side and demand-side factors to facilitate policy action. This also accounts for reviewer 2's suggestion (no 19) to discuss supply-side and demand-side factors in a way to facilitate policy actions.

16. Conclusion: Lines 31 to 38 seem material for the background/introduction.

The concept of task-sharing seems to be appearing for the first time in line 40. It is important to ensure that enough materials exist in the result and discussions to support its appearance in the conclusion. It is however difficult to see these evidence in the current version of the manuscript.

Response: We have now revised the conclusion, removing the reference to task-sharing. Instead we summarise information from the preceding results (ref: pg 15 Strategies for improving surgical delivery) mentioning training and deployment of lower tier staff and surgical assistants in community and primary health centres in Sierra Leone for improving surgical provisioning.

Reviewer: 2

Reviewer Name: James X. Zhang

Institution and Country: The University of Chicago, USA Please state any competing interests or state 'None declared': None declared.

This is an exploratory, qualitative study investigating the determinants of surgical access in a low-income country in West Africa. A better understanding of provider- and patient-perceived factors that impede or facilitate surgical care can lead to the identification of the barriers to an equitable health care system.

17. Conceptualization. While the authors provided a detailed account of experiences of patients' initial decision to seek treatment (Page 8-10), a figure illustrating the key barriers would help conceptualize and streamline the perceived barriers, for example, monetary, belief, organizational, etc, with more detailed breakdown by each category.

Response: On page 11, we have now inserted a figure illustrating the key barriers in terms of affordability and convenience, personal beliefs, and organisational barriers with more detailed breakdown by each category.

18. Treatment pathways. A figure depicting/summarizing the treatment pathways (Page 10-13) can help readers to grasp the challenges in surgical care.

Response: On pg. 13 we have inserted figure 4 summarising patients' main treatment pathways.

19. Strategies. It is unclear what the main lesson is from this study. Supply-side and demand-side factors should be discussed in a way to facilitate policy actions.

Response: As suggested, we have re-ordered the discussion section to consider the main lessons from this study by reviewing the supply-side and demand-side factors to facilitate policy action. This also accounts for reviewer 1's suggestion to have the themes appear at the opening of each paragraph to relate them to the results of this and other studies.

VERSION 2 – REVIEW

REVIEWER	Amos Deogratus Mwaka Makerere University College of Health Sciences, Uganda.
REVIEW RETURNED	30-Sep-2020

GENERAL COMMENTS	Dear Authors, Congratulations for the great improvement in the quality and readability of this important manuscript. The manuscript seeks to expose the issues that influence access to surgical care in Sierra Leone. I appreciate the responses you have provided to the issues raised in the first round of review. However; 1. The abstract, results section still does not bring out the identified themes and sub-themes with their exemplars. It looks like there were two categories of factors that influence access: Demand side and supply side issues. Under each of these categories, there seem to be themes (enablers and barriers) and under each of the themes there are sub-themes for example high cost of care at biomedical facilities (barrier), but similarly a pull factor - affordability and convenient mode of payment to the traditional health practitioners. Both hinder access to the biomedical facilities, but differently. An this is the beauty of your
---

	design - qualitative. It can teas out the issues and place them where each belongs. If the above understanding is the case, then you need to further revise the results section - both in the abstract but also the main manuscript to reflect these issues clearly. For example, on page 10, under results section, there is "Use of traditional medicine". It is not clear what this means. The vice presented from FGD 3, Healers; does not seem to make understanding any better. Your introductory reflection under that theme or sub-theme also does not provide enough guidance to the reader as to where the direction is. For example, is it being said that healers go late or do not access surgical care at biomedical facilities because traditional healers attract them by factor of affordability and more convenient mode of payments? 2. Presentation of results. In general, this section is much improved. But you need to be clear upfront on how you want to present the result: Do you want to present by participants' categories e.g. healers, patients etc, then you develop categories, themes and sub-themes? That is fine but you may end up repeated the themes and sub-themes under each participant categories. The other approach is presenting categories based on findings e.g. demand and supply sides; then the themes and sub-themes are presented, including voices from all categories of participants, comparing and contrasting by the participants' categories. While the health providers said this regarding use of traditional healers, the healers point of views reflect a, b and c, and so on and so on. This is a better approach and is what you are using. You however need to maintain the consistency and comment on each category of participants under every themes and sub-themes. 3. Number of participants in each FGD is really low. You might want to review the literature on FGDs and present a more compelling argument for why you did what you did. You cannot go back to conduct the data collection. For example, when you realized that the numbers were few, why did you not conduct in-depth interviews instead? The interactions and issues that ensue from discussions are critical but do they arise when you have such few numbers as 3? You might want to include citations talking about the recommended size for an FGD, and then argue why you did otherwise; and also include citations to support that position you have taken. 4. Title; The word "Determinant" is synonymous with a quantitative approach where magnitudes of association with accompanying 95% confidence interval indicating statistical significance and non-significance as the case may be. So, in this regard, it might be a good idea to revise the title. 5. Limitations; there is no discussion on limitations under the discussion section. You might want to include this section and discuss detailed issues regarding limitations, with appropriate citations. Regards
--	---

REVIEWER	James X. Zhang The University of Chicago USA
REVIEW RETURNED	12-Oct-2020

GENERAL COMMENTS	Surgical access is critical for the population wellbeing and 5 billion people worldwide lack timely access to safe surgical services. The
---

	study is topically important and may have some interesting implication to healthcare delivery and policy among low and middle-income countries. The utilization of a demand-supply framework helps to elucidate the access barriers. The report discusses a few limitations of the study. This review is focused on the clarity of the report. 1. Significance. The authors stated that “Surgical access is essential for universal health coverage (UHC)”, (Line 34, Page 3) Yet it is unclear what an “universal health coverage” this report refers to. Please clarify. Specifically, for example, the readers will gain a better understanding of the significance of the study if the report can more strongly associate the features of UHC (or a lack of it) with surgical access worldwide. 2. The demand-supply framework. This framework helps to elucidate the challenges in surgical access. One player missing in this framework in the payor. The report did mention that the affordability is a contributing factor for access barrier perceived by the patients. A diagram of illustrating the demand-supply-payor interaction will help the readers to understand better the complex relationship among system-patients-providers. 3. Generalizability. Beyond Sierra Leone, how are the findings and recommendations applicable to other low and middle-income countries particularly those in Africa? The readers will gain greater understanding of the systematic challenges if the authors can elaborate further to state clearly the implication of the study.
--	---

VERSION 2 – AUTHOR RESPONSE

Reviewer: 1

1. The abstract, results section still does not bring out the identified themes and sub-themes with their exemplars. It looks like there were two categories of factors that influence access: Demand side and supply side issues. Under each of these categories, there seem to be themes (enablers and barriers) and under each of the themes there are sub-themes for example high cost of care at biomedical facilities (barrier), but similarly a pull factor - affordability and convenient mode of payment to the traditional health practitioners. Both hinder access to the biomedical facilities, but differently. An this is the beauty of your design - qualitative. It can tease out the issues and place them where each belongs. If the above understanding is the case, then you need to further revise the results section - both in the abstract but also the main manuscript to reflect these issues clearly.

For example, on page 10, under results section, there is "Use of traditional medicine". It is not clear what this means. The vice presented from FGD 3, Healers; does not seem to make understanding any better. Your introductory reflection under that theme or sub-theme also does not provide enough guidance to the reader as to where the direction is. For example, is it being said that healers go late or do not access surgical care at biomedical facilities because traditional healers attract them by factor of affordability and more convenient mode of payments?

Response:

As suggested by Reviewer 1, in order to draw out and clearly indicate the dynamic way in which demand and supply side factors interacted to influence patients' access and use of services, we have revised and reworded both the abstract and the results section. Overall, we have re-ordered the results section (i) to rephrase and order the results more clearly along the two categories of factors that influence access: demand side and supply side issues. In order to do this, we have re-worded the main headings (pg. 9 and pg. 12). (ii) Further, in order to clarify the focus of the subtext under each sub-heading, we have altered the wording of the sub-headings (pg. 10-14) and have placed some text

under different sub-headings to illustrate the direction of the results and what they reveal. (iii) On page 10, we have altered the sub-heading to clarify that the sub-section focuses on 'Pull factors for use of traditional healers'.

2. Presentation of results. In general, this section is much improved. But you need to be clear upfront on how you want to present the result: Do you want to present by participants' categories e.g. healers, patients etc, then you develop categories, themes and sub-themes? That is fine but you may end up repeated the themes and sub-themes under each participant categories.

The other approach is presenting categories based on findings e.g. demand and supply sides; then the themes and sub-themes are presented, including voices from all categories of participants, comparing and contrasting by the participants' categories. While the health providers said this regarding use of traditional healers, the healers point of views reflect a, b and c, and so on and so on. This is a better approach and is what you are using. You however need to maintain the consistency and comment on each category of participants under every themes and sub-themes.

Response: We thank Reviewer 1 for this very useful feedback and have followed the suggested second approach whereby categories are presented based on demand and supply sides. Accordingly, we have re-ordered the results section to rephrase and order the results more clearly along the two categories of factors that influence access: demand side and supply side issues. We then present the themes and sub-themes comparing and contrasting by relevant participants' categories.

3. Number of participants in each FGD is really low. You might want to review the literature on FGDs and present a more compelling argument for why you did what you did. You cannot go back to conduct the data collection. For example, when you realized that the numbers were few, why did you not conduct in-depth interviews instead? The interactions and issues that ensue from discussions are critical but do they arise when you have such few numbers as 3? You might want to include citations talking about the recommended size for an FGD, and then argue why you did otherwise; and also include citations to support that position you have taken.

Response: We have noted the low number of FGD participants as a limitation of our study on page 6 as well as in the discussion section (pg 18). On pg 6, we explain this to be particularly the case for FGDs with traditional healers and SACHOs (3 participants each) which is smaller than average (typically 5-8 participants). We point out however the challenges of recruiting traditional healers who are a particularly hard to reach group within this context and therefore needing to make pragmatic choices with their recruitment We note however that the FGDs were executed through skilled researchers who ensured dialogue and interaction among participants (as distinct from a group interview), an important characteristic of FGDs. We also explain that within existing time and budget constraints, we tried to ensure that FGDs represented a broad spectrum of voices and identify specific concerns of particular groups. On pg. 16, we provide relevant citations to support the position we have taken.

4. Title; The word "Determinant" is synonymous with a quantitative approach where magnitudes of association with accompanying 95% confidence interval indicating statistical significance and non-significance as the case may be. So, in this regard, it might be a good idea to revise the title.

Response: We have removed the word 'determines' and replaced it with 'factors shape'. The title now reads: 'What factors shape surgical access in West Africa? A qualitative study exploring patient and provider experiences of managing injuries in Sierra Leone'.

5. Limitations; there is no discussion on limitations under the discussion section. You might want to include this section and discuss detailed issues regarding limitations, with appropriate citations.

Response: On pg. 16 we have inserted a paragraph discussing the limitation of the study and included relevant citations. Bearing in mind, Reviewer 1's earlier comments about the size of the FGDs we have highlighted this as a particular limitation of this study, inserting relevant citations to explain the possible issues regarding limitations.

Reviewer: 2

1. Significance. The authors stated that "Surgical access is essential for universal health coverage (UHC)", (Line 34, Page 3) Yet it is unclear what an "universal health coverage" this report refers to.

Please clarify. Specifically, for example, the readers will gain a better understanding of the significance of the study if the report can more strongly associate the features of UHC (or a lack of it) with surgical access worldwide.

Response: On page 3, we have inserted a clearer definition of the UHC and added in the citation from the World Health Organization document that is referred to.

2. The demand-supply framework. This framework helps to elucidate the challenges in surgical access. One player missing in this framework in the payor. The report did mention that the affordability is a contributing factor for access barrier perceived by the patients. A diagram of illustrating the demand-supply-payor interaction will help the readers to understand better the complex relationship among system-patients-providers.

Response: We have inserted a diagram, Figure 3 on page 18 illustrating the demand-supply-payer interaction to help readers understand better the complex relationship among system-patients-providers.

3. Generalizability. Beyond Sierra Leone, how are the findings and recommendations applicable to other low and middle-income countries particularly those in Africa? The readers will gain greater understanding of the systematic challenges if the authors can elaborate further to state clearly the implication of the study.

Response: On page 18, we have inserted text reflecting on the generalizability of the results beyond Sierra Leone and included recommendations for improving surgical care in other low resource settings. Accordingly, we highlight key lessons from this study which will be applicable to other contexts. This includes, on the demand side, financial mechanisms for ensuring more affordable care for patients as well as systematic community outreach to improve awareness about early care seeking. On the service delivery side, our results highlight the importance of continuing professional development for incentivising surgical providers and strengthening communication systems between facilities to reduce treatment delays. In addition, we have added information on page 16 to highlight the relevance of surgical innovations in low-resource settings citing from appropriate sources.

VERSION 3 – REVIEW

REVIEWER	Amos Deogratius Mwaka Makerere University
REVIEW RETURNED	01-Dec-2020

GENERAL COMMENTS	Dear Authors, Thank you for the great improvement on the manuscript. I have enjoyed reading through it this time. 1. I strongly recommend you tighten on the ideas and the messages a little more - throughout the manuscript. Some sections lack clear messages. For example, the conclusion in the main text leaves me with very scanty message that are derived from the results. I do not get the message in this sentence for instance: "Strong policy support will need to be backed up with supportive legislation, integrating surgical care into national planning exercises and accompanying resource commitment for ongoing research and health system strengthening". What is the message here? What does strong policy support mean? Support for what? You might want to be more specific to what your main findings mean in terms of policy, practice and research in the country and beyond. I do not see this coming out clearly in the current version of the manuscript. 2. Limitations and strength
---

	The limitations are not explained. For example, it is said how "Two data collectors were practicing doctors in Sierra Leone which may have influenced some interactions with respondents" is a limitation. Also, generalizability often applies for quantitative studies. In qualitative, we talk more of transferability of the findings. 3. Table 1 (b) could be made better. What are - anesthetist traditional healers etc.? That answer to what they are is their label to be written on the table. I wish you the very best with this manuscript. Regards A.D Mwaka
--	---

REVIEWER	James X. Zhang The University of Chicago
REVIEW RETURNED	17-Dec-2020

GENERAL COMMENTS	The authors have been responsive to the reviewers' comments and suggestions; the paper has been improved to be in the shape for publication.
--

VERSION 3 – AUTHOR RESPONSE

Reviewer: 1

1. I strongly recommend you tighten on the ideas and the messages a little more - throughout the manuscript. Some sections lack clear messages. For example, the conclusion in the main text leaves me with very scanty message that are derived from the results. I do not get the message in this sentence for instance: "Strong policy support will need to be backed up with supportive legislation, integrating surgical care into national planning exercises and accompanying resource commitment for ongoing research and health system strengthening". What is the message here? What does strong policy support mean? Support for what?

You might want to be more specific to what your main findings mean in terms of policy, practice and research in the country and beyond. I do not see this coming out clearly in the current version of the manuscript.

RESPONSE: Thanks for the comment. We have made a clarification for the specific example that you mention wherein on page 21, we have modified the text as suggested to specify the policy action needed in terms of allocating resources and integrating surgery into national planning processes. However, otherwise it is difficult for us to determine the specifics around "tightening up the messages" and note that the other reviewer as well as all co-authors are satisfied with the quality of the manuscript and feel that it is clear.

2. Limitations and strength

The limitations are not explained. For example, it is said how "Two data collectors were practicing doctors in Sierra Leone which may have influenced some interactions with respondents" is a limitation.

RESPONSE: We have inserted text (page 3) to clarify that "Two data collectors were practicing doctors in Sierra Leone. Practicing doctors might be seen as authority figures making ancillary staff and patients less likely to answer critically, given perceived power differentials in such relationships."

3. Also, generalizability often applies for quantitative studies. In qualitative, we talk more of transferability of the findings.

RESPONSE: In the summary box (page 3), we have replaced the word 'generalisability' with

'transferability of the results to other contexts'.

4. Table 1 (b) could be made better. What are - anaesthetist traditional healers etc.? That answer to what they are is their label to be written on the table.

RESPONSE: We have added the word ' nurse' to clarify the definition of anaesthetist and further inserted a note below table 1(b) to define nurse anaesthetist and traditional healers.(pages 7,8)

Reviewer: 2

Comments to the Author:

The authors have been responsive to the reviewers' comments and suggestions; the paper has been improved to be in the shape for publication.

RESPONSE: noted with thanks.

Reviewer: 1

Competing interests of Reviewer: I have no competing interests

Reviewer: 2

Competing interests of Reviewer: None declared.